# Experimental Study on Optimization of Phosphogypsum Suspension Decomposition Conditions under Double Catalysis

**DOI:** 10.3390/ma14051120

**Published:** 2021-02-27

**Authors:** Pinjing Xu, Hui Li, Yanxin Chen

**Affiliations:** College of Materials Science and Engineering, Xi’an University of Architecture and Technology, Xi’an 710055, China; sunshineli@vip.sina.com (H.L.); yx_ch@126.com (Y.C.)

**Keywords:** phosphogypsum, suspended state decomposition, decomposition rate, desulfurization rate

## Abstract

Phosphogypsum (PG) is not only a solid waste discharged from the phosphate fertilizer industry, but also a valuable resource. After high-temperature heat treatment, it can be decomposed into SO_2_ and CaO; the former can be used to produce sulfuric acid, and the latter can be used as building materials. In this paper, the catalytic thermal decomposition conditions of phosphogypsum were optimized, and the effects of the reaction temperature, reaction atmosphere, reaction time and carbon powder content on the decomposition of phosphogypsum were studied. The research shows that the synergistic effect of carbon powder and CO reducing atmosphere can effectively reduce the decomposition temperature of phosphogypsum. According to the results of the orthogonal test under simulated suspended laboratory conditions, the factors affecting the decomposition rate of phosphogypsum are temperature, time, atmosphere and carbon powder content in turn, and the factors affecting the desulfurization rate are time, temperature, atmosphere and carbon powder content in turn. Under laboratory conditions, the highest decomposition rate and desulfurization rate of phosphogypsum are 97.73% and 97.2%, and the corresponding reaction conditions are as follows: calcination temperature is 1180 °C, calcination time is 15 min, carbon powder content is 4%, and CO concentration is 6%. The results of thermal analysis of phosphogypsum at different temperature rising rates show that the higher the temperature rising rate, the higher the initial temperature of decomposition reaction and the temperature of maximum thermal decomposition rate, but the increase in the temperature rising rate will not reduce the decomposition rate of phosphogypsum.

## 1. Introduction

Phosphogypsum (PG) is a solid waste produced during the wet production of phosphoric acid in the phosphate fertilizer industry, and its main component is CaSO_4_·2H_2_O or CaSO_4_·0.5H_2_O [1,2,3]. In the wet phosphoric acid production process, about 5–6 tons of phosphogypsum are discharged for every ton of phosphoric acid produced [4]. According to incomplete statistics, there are more than 100 million tons of phosphogypsum slag piled up in China at present, and about 30 million tons of phosphogypsum slag will be newly produced every year, but its comprehensive utilization rate is only about 30% [5,6]. A huge amount of phosphogypsum has not been effectively utilized. Stockpiling and burying these phosphogypsum residues not only occupies a large amount of land, but also easily causes serious pollution to the atmosphere and groundwater. Environmental protection pressure caused by phosphogypsum discharge has become one of the important restrictive factors for the development of the phosphate fertilizer industry [7,8,9]. Phosphogypsum is not only an industrial waste, but also a valuable resource. Phosphogypsum can be decomposed into SO_2_ and CaO after heat treatment at high temperature. The former can be used to produce sulfuric acid, while the latter can be used as raw material of building materials. On the one hand, this process can eliminate the environmental pollution caused by phosphogypsum, turn waste into wealth, and solve the problem of a serious shortage of sulfur resources in China; on the other hand, it can save resources by using the generated lime slag as the raw material for the production of building materials [10,11,12,13,14]. However, the theoretical decomposition temperature of phosphogypsum is as high as 1700 °C, and the high energy consumption of phosphogypsum thermal decomposition and the instability of decomposition product components have not been completely solved, which greatly restrict the industrial application of phosphogypsum to produce sulfuric acid and lime [15,16,17,18,19].

At present, scholars’ research on phosphogypsum pyrolysis mostly focuses on gas as a catalyst or solids as a catalyst [20,21,22,23,24,25,26]. Okumura [27] studied the influence of the CO–CO_2_ concentration ratio on calcium sulfate decomposition in CO-CO_2_-N_2_ atmosphere and found that the regeneration of Cao from CaSO_4_ was strongly dependent on the CO/CO_2_ concentration ratio. In the aspect of gas–solid catalytic reactions, Xiao [28] studied the decomposition of CaSO_4_ in four different atmospheres: high purity N_2_, pu897re CO_2_, 9% O_2_ + 91% N_2_, 10% O_2_ and 90% CO_2_ under the same temperature control procedure. Gruncharov [29] studied the decomposition of phosphogypsum in a H_2_-CO_2_-H_2_O-Ar mixed atmosphere by thermogravimetry. In the aspect of solid–solid catalytic reactions, Zheng [30] studied the solid–solid reaction between phosphogypsum and high-sulfur coal in a tubular electric furnace under N_2_ atmosphere. Merwe [31] studied the change of the decomposition temperature of phosphogypsum in the presence of ZnO and Fe_2_O_3_ alone or in combination at different heating rates.

At present, some progress has been made in the study of phosphogypsum catalytic decomposition, but most of the studies focus on the single reaction mechanism in gas–solid catalytic reactions or solid–solid catalytic reactions, and the ideal decomposition rate and desulfurization rate of phosphogypsum have not been obtained. There are few studies on reducing the reaction temperature of phosphogypsum by the combined action of atmosphere and external catalyst. Therefore, this study aims at reducing the decomposition temperature of phosphogypsum, improving the decomposition rate and desulfurization rate of phosphogypsum, and based on the combined action of gas–solid catalytic reaction and solid–solid catalytic reaction mechanism, using a synchronous thermal analyzer and self-developed air-blown suspension furnace to study the catalytic decomposition of phosphogypsum in a simulated suspension state. By adding carbon powder into phosphogypsum and adjusting the reaction temperature, time and atmosphere, the decomposition conditions of phosphogypsum were explored and optimized, and the optimal reaction conditions of phosphogypsum under laboratory conditions were analyzed, which provided theoretical and data support for the development of new technology and new equipment for phosphogypsum decomposition.

## 2. Experiment

### 2.1. Materials and Reagents

The average particle size of phosphogypsum (A phosphorus chemical Group Co., Ltd., Guizhou, China) used in the experiment was 52.7 μm, which was determined by a laser particle size distribution instrument. Due to the high humidity of raw materials and a large amount of caking, the materials were dehydrated and dried in suspension at about 180 °C. The chemical composition is shown in Table 1, the mineral composition is shown in Figure 1, the particle size distribution is shown in Figure 2, and the SEM of raw materials is shown in Figure 3. According to Figure 1, the main mineral of phosphogypsum is CaSO_4_·0.5H_2_O, and in addition, it also contains a small amount of SiO_2_ impurities. According to Figure 2, the main particle size range of phosphogypsum raw material is between 70–200 μm, and the content is 53.61%; the content of particles smaller than 70 μm is 37.49%; the content of particles with a particle size above 200 μm is 8.9%. It can be concluded from Figure 3 that phosphogypsum in the sample mainly exists in the form of plate or strip aggregates, with a regular crystal form and good crystallinity. Most aggregates are attached with fine plate-like particles. Relatively speaking, the crystallinity of the particles at the edge of the aggregates is better than that of the inner particles.

Other materials used in the study were ethylene glycol, ethanol, anhydrous ethanol of benzoic acid, potassium iodate standard titration solution and sodium thiosulfate standard titration solution (A Technology Co., Ltd., Xi’an, China).

### 2.2. Mechanism of Experiment

There are three possible chemical reactions in which CaSO_4_ decomposition, and its decomposition products may participate [32,33,34].

(1)The equation of direct thermal decomposition reaction of CaSO_4_ at high temperature is:

(1)$$2CaSO_{4}\to 2CaO+2SO_{2}(↑)+O_{2}(↑)$$

(2)Gas–solid reaction mechanism: According to the theory, when reducing a gas medium such as CO and it comes into contact with solid CaSO_4_ at proper high temperatures, not only heat and mass transfer occurs between gas and solid, but also the following decomposition reaction takes place.

Main reaction:(2)$$CaSO_{4}+CO\to CaO+SO_{2}(↑)+CO_{2}(↑)$$

Side reactions:(3)$$CaSO_{4}+4CO\to CaS+4CO_{2}(↑)$$
(4)$$CaS+\frac{3}{2}O_{2}\to CaO+SO_{2}(↑)$$
(5)$$CaS+2O_{2}\to CaSO_{4}$$

(3)Solid–solid reaction mechanism: According to this theory, the coexistence of gypsum and carbon can promote decomposition, and its reaction is:

Main reaction:(6)$$2CaSO_{4}+C\to 2CaO+2SO_{2}(↑)+CO_{2}(↑)$$

Side reactions:(7)$$CaSO_{4}+4C\to CaS+4CO(↑)$$
(8)$$CaSO_{4}+3C\to CaS+CO_{2}(↑)+2CO(↑)$$

According to the results of the previous thermodynamic calculation, the reaction temperature required for the decomposition of CaSO_4_ itself is very high, which is not easy to obtain. Generally, the decomposition of CaSO_4_ is promoted by adding a reducing agent. Therefore, this paper adopts the combination of gas–solid reaction mechanisms and solid–solid reaction mechanisms to reduce the decomposition temperature of CaSO_4_, that is, adding carbon powder and introducing CO gas to adjust the reaction atmosphere. Table 2 shows the starting temperature and free energy of possible chemical reactions in phosphogypsum decomposition.

### 2.3. Experimental Device

The schematic diagram of the small air-blown suspension furnace used in this paper is shown in Figure 4. In order to simulate the suspension and dispersion calcination of phosphogypsum raw materials in airflow, a box with a screen at the bottom was made. The phosphogypsum sample was laid flat on the screen at the bottom of the box, and the thickness of the material layer is about 2 mm. The gases CO, CO_2_ and N_2_ used in the experiment passed through the metering device in proportion and then passed through the material layer evenly. The sample residue obtained after the reaction was taken out from the top of the furnace with the aid of the box.

In the study, the chemical composition of phosphogypsum was detected by an S4 PIONEER X-ray fluorescence spectrometer analyzer (Bruker Corporation, Karlsruhe, Baden-Württemberg, Germany), the mineral composition of phosphogypsum and decomposed residue was detected by a D–MAX/2500 X-ray diractometer (Rigaku Corporation, Zhaodao, Tokyo, Japan), the thermal decomposition process of phosphogypsum was analyzed by an STA409PC thermal analyzer (NETZSCH Corporation, Selbu, Germany), CaO in the residue was detected by a CA-5 free calcium rapid analyzer (Shanghai leao test instrument Co., Ltd, Shanghai, China), the microscopic morphology was observed with a Quanta 2000B scanning electron microscope (FEI Corporation, Hillsboro, OR, USA), and the particle size distribution of raw materials was determined by a HELOS-RODOS laser particle size distribution instrument (SYMPATEC GmbH Corporation, Clausthal-Zellerfeld, Germany). 

### 2.4. Characterization of Results

Because the main and side reactions of the phosphogypsum decomposition process are complex, the decomposition rate (*ϕ*) and desulfurization rate (*Ψ*) are often used to characterize the degree of the phosphogypsum decomposition reaction.

Decomposition rate (*ϕ*): The percentage of CaSO_4_ in the sample that has been decomposed into CaO and CaS in the main and side reaction products to the total amount of CaSO4, and the equation is as follows:(9)$$\phi =\frac{M_{CaSO_{4}}^{CaO}+M_{CaSO_{4}}^{CaS}}{M_{CaSO_{4}}^{CaO}+M_{CaSO_{4}}^{CaS}+M_{CaSO_{4}}^{CaSO_{4}}}\times 100\%$$

Desulfurization rate (*Ψ*): The percentage of S removed from CaSO_4_ in the sample, and the equation is as follows:(10)$$\Psi =\frac{M_{CaSO_{4}}^{CaO}}{M_{CaSO_{4}}^{CaO}+M_{CaSO_{4}}^{CaS}+M_{CaSO_{4}}^{CaSO_{4}}}\times 100\%$$

In the equation: $M_{CaSO_{4}}^{CaO}$ means that the CaO content measured in the residue is converted into CaSO_4_ content in the raw material through the calculation of the chemical reaction equation; $M_{CaSO_{4}}^{CaS}$ means that the CaS content measured in the residue is converted into CaSO4 content in the raw material through the calculation of the chemical reaction equation; $M_{CaSO_{4}}^{CaS}$ is the undecomposed CaSO4 content measured in the residual residue after decomposition.

In order to calculate the decomposition rate (*ϕ*) and desulfurization rate (*Ψ*) of the phosphogypsum decomposition reaction, it is necessary to analyze and detect CaO, CaS and total S in the residue. In which:(1)CaO in residue was detected by a free calcium rapid analyzer. An amount of 0.4 g of the sample was weighed (depending on the content of free calcium oxide), put into a dry 250 mL conical flask, then 15–20 mL of ethylene glycol-ethanol solution was added, and the conical flask was gently shaken to disperse the sample (the temperature was 100–110 °C). The small condenser tube was placed on the free calcium analyzer, the power switch was turned on, and the working time to was adjusted to 3 min after the circulating pump worked normally. The solution was stirred at a lower speed and the temperature was raised at the same time. When the condensed ethanol started dripping, the start button was pressed to start timing, the temperature voltmeter was adjusted to about 150 V, and the rotating speed was increased. After timing and extraction, the conical flask taken off, it was titrated with benzoic acid absolute ethyl alcohol standard solution until the red color disappeared and the volume was recorded. According to the consumed volume of benzoic acid absolute ethyl alcohol standard solution and titre of calcium oxide, the percentage content of free calcium oxide in the sample was obtained.
(11)$$f_{CaO}\%=\frac{T_{CaO}\times V\times 100}{G\times 1000}$$

In the formula, *T_CaO_* is the milligram equivalent to calcium oxide per milliliter of benzoic acid absolute ethyl alcohol standard solution (mg/mL); *V* is the volume of benzoic acid absolute ethyl alcohol standard solution consumed by titration (mL); *G* is the weight of the residue sample (g).

(2)The determination of CaS in the residue is carried out according to the National Standard for chemical analysis methods of cement (GB/T176-2017).(3)The total sulfur S in the residue is detected by the fluorescence spectrometer.

## 3. Results and discussion

### 3.1. Thermal Analysis of Phosphogypsum Decomposition Process

In order to explore the influence of experimental conditions on the thermal decomposition reaction process of phosphogypsum, the reaction process analysis of phosphogypsum samples under two experimental conditions of self-decomposition, adding carbon powder into phosphogypsum and introducing CO atmosphere was carried out on a synchronous thermal analyzer with the same rate of temperature rise (10 °C/min). The test results are shown in Figure 5a,b.

According to the analysis in Figure 5a, the thermal decomposition process of phosphogypsum can be divided into two stages under inert protective atmosphere and a temperature rising rate of 10 °C/min:(1)Dehydration stage of phosphogypsum. The dehydration process ends at 180 °C. At this stage, the temperature rises after the phosphogypsum absorbs heat, which leads to the physical process of dehydration of phosphogypsum (calcium sulfate hemihydrate), and the weight loss rate accounts for about 3.93% of the raw materials.(2)The decomposition stage of phosphogypsum. The decomposition reaction of phosphogypsum starts from 1228.8 °C, which is the main reaction stage of pyrolysis. Here, the chemical reaction shown in Equation (1) mainly occurs, that is, the self-decomposition reaction of phosphogypsum. At this stage, phosphogypsum is pyrolyzed to produce CaO and SO_2_ gas, resulting in obvious weight loss. The weight loss rate reaches the maximum at 1282 °C, and the weight loss rate is 42.8%. Until the temperature reaches the upper operating temperature of 1400 °C, the main reaction is still not completely finished.

According to the analysis in Figure 5b, the thermal decomposition process of phosphogypsum can also be divided into two stages when 4% carbon powder is added into phosphogypsum and CO gas is introduced at a temperature rising rate of 10 °C/min:(1)Dehydration stage of phosphogypsum. Similar to Figure 5a, this stage ends at about 180 °C, which is mainly the dehydration process of phosphogypsum, and the weight loss rate accounts for about 4.40% of the raw materials.(2)The decomposition stage of phosphogypsum. The thermal decomposition reaction of phosphogypsum started at 907.7 °C, and its weight loss rate reached the maximum at 963.9 °C. The main reaction here is the result of chemical Equations (2) and (6), that is, the gas–solid catalytic reaction and the solid–solid catalytic reaction. Compared with Figure 5a, the initial temperature of the main decomposition reaction and the maximum temperature of the weight loss rate both decreased by about 320 °C, and the weight loss rate in this main reaction stage was 37.3%. Because of the interference of heat release from the combustion of carbon powder, its DSC (Differential Scanning Calorimetry)endothermic curve is not obvious.

Comparing Figure 5a,b, it can be seen that the decomposition temperature of phosphogypsum in inert atmosphere is above 1200 °C at the same temperature rising rate. However, when 4% carbon powder is added into the sample and CO-reducing atmosphere is introduced, the initial temperature of the phosphogypsum decomposition reaction and the corresponding temperature of the fastest decomposition rate are greatly reduced, and the initial temperature of the decomposition reaction is reduced to below 1000 °C, indicating that the decomposition temperature of phosphogypsum can be effectively reduced under the synergistic action of carbon powder and reducing atmosphere. The weight loss rate of the decomposition reaction under two conditions has little change, which is about 40%, which is about 70% of the theoretical weight loss rate.

### 3.2. Synergistic Effect of Multiple Factors on Suspended Decomposition of Phosphogypsum

Based on the thermal analysis results of phosphogypsum decomposition process in Figure 5, in order to study the thermal decomposition law of phosphogypsum under the synergistic effect of multiple factors, this paper designed an orthogonal test for factors such as reaction temperature, reaction time, reaction atmosphere and carbon powder content, and carried out systematic research. The decomposition rate (*ϕ*) and desulfurization rate (*Ψ*) of suspended phosphogypsum under different atmosphere conditions were tested by using the air-blown suspension furnace shown in Figure 4. The orthogonal test scheme is shown in Table 3, the atmosphere conditions are shown in Table 4, and the results of orthogonal test and range analysis are shown in Table 5 and Table 6.

In Table 6, K1, K2 and K3 are the sum of the decomposition rate or desulfurization rate of level 1, level 2 and level 3 in orthogonal test, and R is the extreme difference of each factor. It can be concluded from the test results that:

(1) The decomposition conditions that make the decomposition rate and desulfurization rate of phosphogypsum achieve better results at the same time are the same, that is, the reaction temperature is 1180 °C; the reaction time is 30 min; the carbon powder content is 4%; the reaction atmosphere: Pco/Pco_2_ is 0.22 (partial pressure ratio of CO and CO_2_), the concentration of CO is 6%.

(2) In the horizontal range of the orthogonal test, the order of factors affecting decomposition rate is temperature > time > atmosphere > carbon powder content. However, the range numbers of the reaction temperature and time are 47.81 and 42.75, respectively, and there is no obvious difference between the temperature and time on the decomposition rate of phosphogypsum. When the temperature is 1120–1180 °C and the time is 20–30 min, the influence of the level change on the decomposition rate is not very great, and the decomposition rate of phosphogypsum does not increase much.

(3) The order of factors affecting desulfurization rate is time > temperature > atmosphere > carbon powder content. The range numbers of time and temperature are 30.74 and 28.11, respectively, and there is no obvious difference between the time and temperature on the desulfurization rate of phosphogypsum. When the temperature is between 1120 °C and 1180 °C and the time is between 20 and 30 min, the influence of the level change on the desulfurization rate of phosphogypsum is less than that on the decomposition rate, and the improvement of desulfurization rate is very small.

### 3.3. Effect of Different Atmosphere Conditions on Thermal Decomposition Reaction of Phosphogypsum in Suspension State

According to the results of the orthogonal test, under the condition of a relatively fixed reaction temperature and time, the influence of atmosphere conditions on the decomposition rate (*ϕ*) and desulfurization rate (*Ψ*) of phosphogypsum is particularly important. The atmosphere factor is influenced by the percentage of CO and the partial pressure ratio of CO and CO_2_ (Pco/Pco_2_). Therefore, this paper also studied the influence of the change of atmosphere on the thermal decomposition of phosphogypsum in suspension under the conditions of a fixed reaction temperature (1180 °C), different reaction time (15 min, 20 min) and different carbon powder content (4%, 5%). The different atmosphere conditions are shown in Table 7, and the test results are shown in Figure 6 and Figure 7.

Figure 6 shows that the best decomposition rate of phosphogypsum reaches 97.73%, and the corresponding reaction conditions are as follows: reaction temperature 1180 °C, carbon powder addition 4%, and reaction time 15min. The reaction atmosphere contains 6% CO, 30% CO_2_, 64% N_2_ and Pco/$P_{{\mathrm{CO}}_{2}}$ is 0.2.

According to the analysis in Figure 7, the optimal desulfurization rate of phosphogypsum reaches 97.20%, and the corresponding reaction conditions are consistent with those when phosphogypsum reaches the optimal decomposition rate. The experimental results under different atmosphere conditions show that the decomposition rate and desulfurization rate of phosphogypsum have reached more than 90% when the reaction temperature is 1180 °C, the reaction time is 15–20 min, the content of carbon powder is 4.0–5.0%, the concentration of CO in the atmosphere is 5.0–6.0%, and the Pco/Pco_2_ is 0.18–0.20. The optimum test results: the decomposition rate is 97.73%, and the desulfurization rate is 97.20%.

Figure 8 is a comparative analysis of mineral composition of phosphogypsum raw material and decomposed residue. In the test conditions, the calcination temperature is 1180 °C, calcination time is 15min, carbon powder content is 4%, CO concentration is 6%, and Pco/Pco_2_ is 0.2. It can be concluded that CaSO_4_·0.5H_2_O mineral phase does not exist in the residue after calcination and decomposition of phosphogypsum, and the decomposition is relatively complete. The main mineral phase in the residue is CaO, followed by SiO_2_, which indicates that the SiO_2_ impurities in phosphogypsum did not undergo a phase change during calcination. In addition, the material also contains a small amount of Ca(OH)_2_, which is mainly due to the reaction of CaO with the moisture in the air during storage.

### 3.4. Reaction Mechanism of Suspended Decomposition of Phosphogypsum

In order to analyze the role of solid–gas reactions and solid–solid reactions in the thermal decomposition of phosphogypsum, an experimental study was carried out in an air-blown suspension furnace at a fixed temperature (1180 °C) and a fixed carbon powder content (4%), and the decomposition rate and desulfurization rate of phosphogypsum under different atmosphere conditions were tested. The experimental atmosphere conditions are shown in Table 8, and the experimental results are shown in Figure 9.

According to Figure 9, when CO does not exist in the mixed gas (S-1), the decomposition rate and desulfurization rate of phosphogypsum are 4.39% and 4.10%, respectively. When the amount of CO in the gas is 5% (S-3), the decomposition rate and desulfurization rate of phosphogypsum are 96.62% and 96.10%, respectively. This shows that if only carbon powder is added as a reducing agent in the reaction process, only solid–solid reactions will occur, which is not conducive to the decomposition reaction of phosphogypsum. Therefore, both gas and solid reducing agents must be added at the same time to promote the reaction. This shows that the solid–gas and solid–solid reaction mechanisms play a corresponding role in the decomposition process of phosphogypsum.

### 3.5. Effect of Temperature Rising Rate on Decomposition Reaction of Phosphogypsum

In order to study the effect of temperature rising rate on the decomposition reaction of phosphogypsum, the phosphogypsum with 4% carbon content was analyzed by TG-DSC at four heating rates (10 °C/min, 15 °C/min, 30 °C/min, 45 °C/min). The reaction atmosphere adopted the optimum conditions in the above-mentioned phosphogypsum suspension decomposition test, with CO 6% and Pco/Pco_2_ 0.2. The test results are shown in Figure 10a–d respectively.

It can be seen from Figure 10 that under different temperature rising rates, the maximum pyrolysis rate in the DTG curve shows an increasing trend corresponding to the temperature, and the main reaction interval also increases slightly. When the temperature rising rate is 10 °C/min, the maximum decomposition rate corresponds to the lowest temperature, which is 963.9 °C. When the temperature rising rate is 45 °C/min, the maximum decomposition rate corresponds to the highest temperature, which is 1138.6 °C, with a difference of nearly 150 °C. This is because when the temperature reaches the same temperature, the higher the rate of temperature rise, and the shorter the reaction time of the sample, while the concentration of CO-reducing gas is limited, so some substances are too late to react, and the reaction rate and degree decrease, which leads to thermal hysteresis and leads to the curve moving to high temperature measurement. When the air-blown suspension furnace is used for the experiment, the phosphogypsum material is directly put into the furnace at a high temperature, that is, the temperature rise is an instantaneous process, and the temperature rise rate is very high. Therefore, the pyrolysis test of phosphogypsum in the air-blown suspension furnace is closer to the thermal analysis test when the temperature rises at 45 °C/min, and the two decomposition temperatures are 1180 °C and 1138.6 °C, which are very close to each other. According to the results of the thermal analysis test, the weight loss rate of phosphogypsum under four temperature rising rates is about 40%, and the increase in the temperature rising rate has no effect on the final complete decomposition of phosphogypsum. Therefore, it can be expected that if the industrial suspension calcination of phosphogypsum is realized and the heat and mass transfer rate is increased, phosphogypsum can undergo a complete decomposition reaction in a very short time.

## 4. Conclusions

In this paper, the thermal decomposition test of phosphogypsum was carried out by using a synchronous thermal analyzer and self-developed air-blown suspension furnace. The main research conclusions are as follows:

(1) The results of the thermal analysis at the same temperature rising rate show that adding carbon powder and introducing CO-reducing atmosphere into phosphogypsum can effectively reduce the decomposition temperature of phosphogypsum, and the initial temperature of decomposition can be reduced from 1228.8 °C to 907.7 °C. The results of the thermal analysis of phosphogypsum at different temperature rising rates show that the higher the temperature rising rate is, the higher the initial temperature of decomposition reaction and the temperature of maximum thermal decomposition rate are. The maximum decomposition rate is 1138.6 °C when the temperature rising rate is 45 °C/min, and the temperature rising rate has little effect on the decomposition rate of phosphogypsum.

(2) The simulated suspension calcination experiment in a small air-blown suspension furnace shows that the factors affecting the decomposition of phosphogypsum include the reaction temperature, reaction time, CO concentration, Pco/Pco_2_ and carbon powder content. The factors affecting the decomposition rate are the temperature, time, atmosphere and carbon powder content from primary to secondary, while the factors affecting desulfurization rate are the time, temperature, atmosphere and carbon powder content from primary to secondary. The synergistic effect of carbon powder and CO-CO_2_-N_2_ atmosphere can effectively reduce the decomposition temperature of phosphogypsum.

(3) Under laboratory conditions, the highest decomposition rate and desulfurization rate of phosphogypsum are 97.73% and 97.2%, and the corresponding reaction conditions are as follows: the calcination temperature is 1180 °C, calcination time is 15min, carbon powder content is 4%, CO concentration is 6%, and Pco/Pco_2_ is 0.2. It can be expected that the reaction time will be greatly shortened without reducing the decomposition rate and desulfurization rate of phosphogypsum.

## Figures and Tables

**Figure 1 materials-14-01120-f001:**
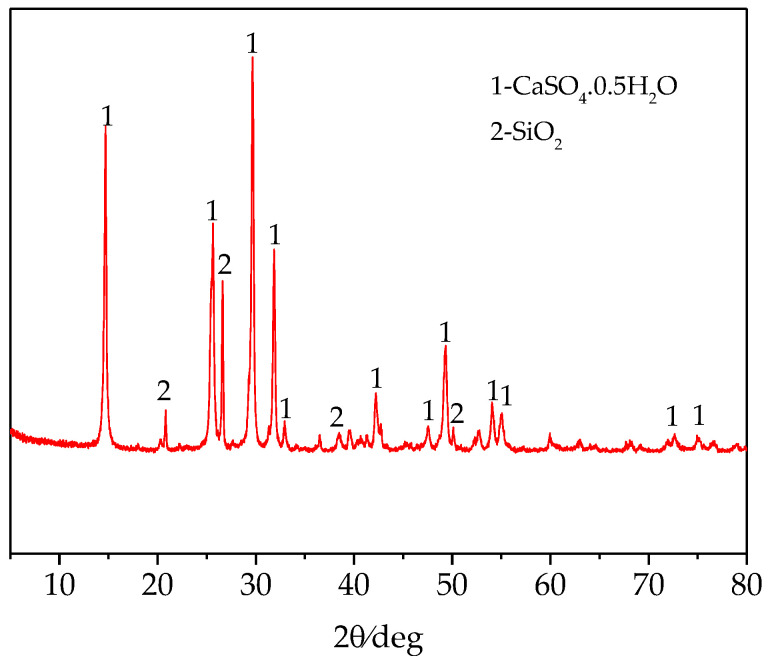
XRD test results of phosphogypsum.

**Figure 2 materials-14-01120-f002:**
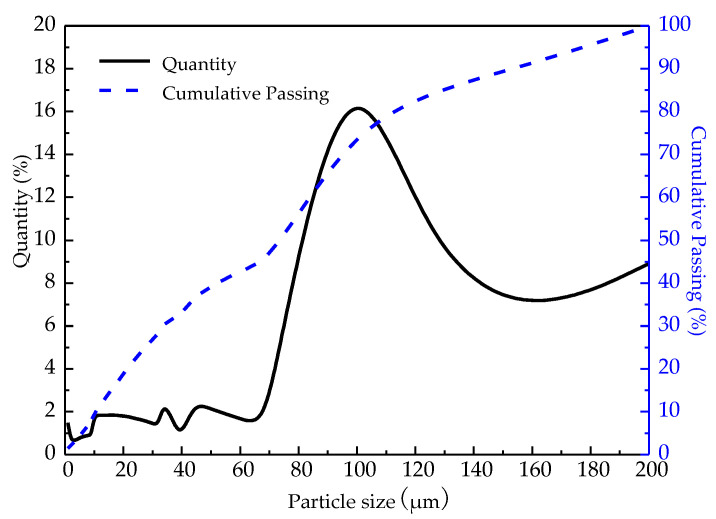
Particle size distribution of phosphogypsum raw materials.

**Figure 3 materials-14-01120-f003:**
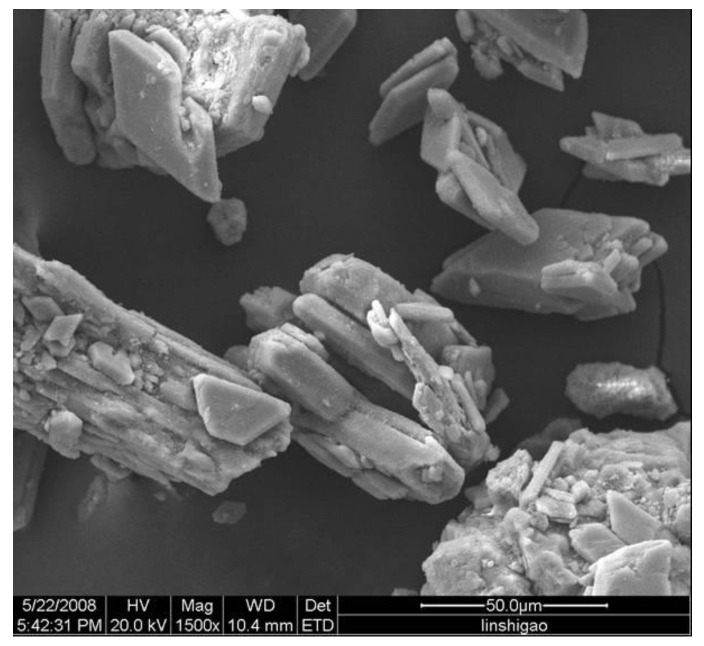
SEM of phosphogypsum raw material.

**Figure 4 materials-14-01120-f004:**
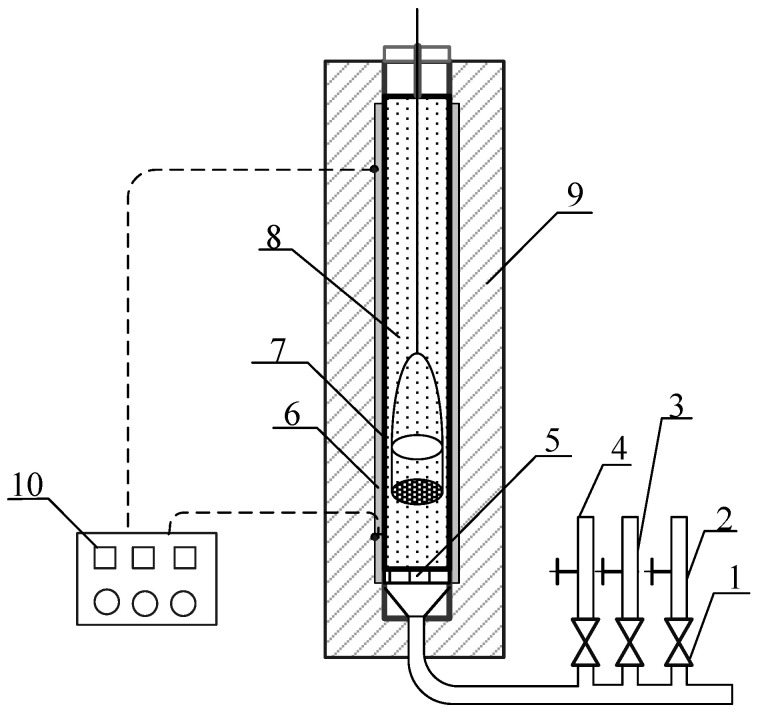
Gas blowing suspension furnace for phosphogypsum decomposition: (**1**) glass rotameter; (**2**) CO gas; (**3**) O_2_ gas; (**4**) N_2_ gas; (**5**) air distribution plate; (**6**) corundum pipe; (**7**) silicon carbon rod; (**8**) material box with screen at the bottom; (**9**) insulation layer; (**10**) automatic temperature control instrument.

**Figure 5 materials-14-01120-f005:**
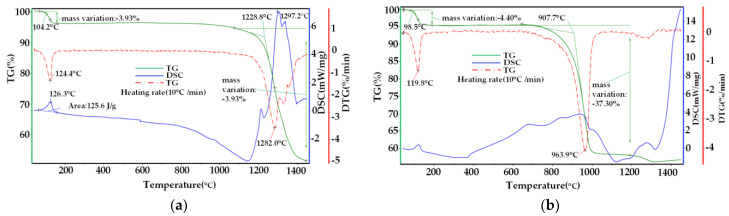
Thermal analysis of phosphogypsum decomposition: (**a**) Phosphogypsum decomposes itself; (**b**) adding 4% carbon powder while introducing CO atmosphere.

**Figure 6 materials-14-01120-f006:**
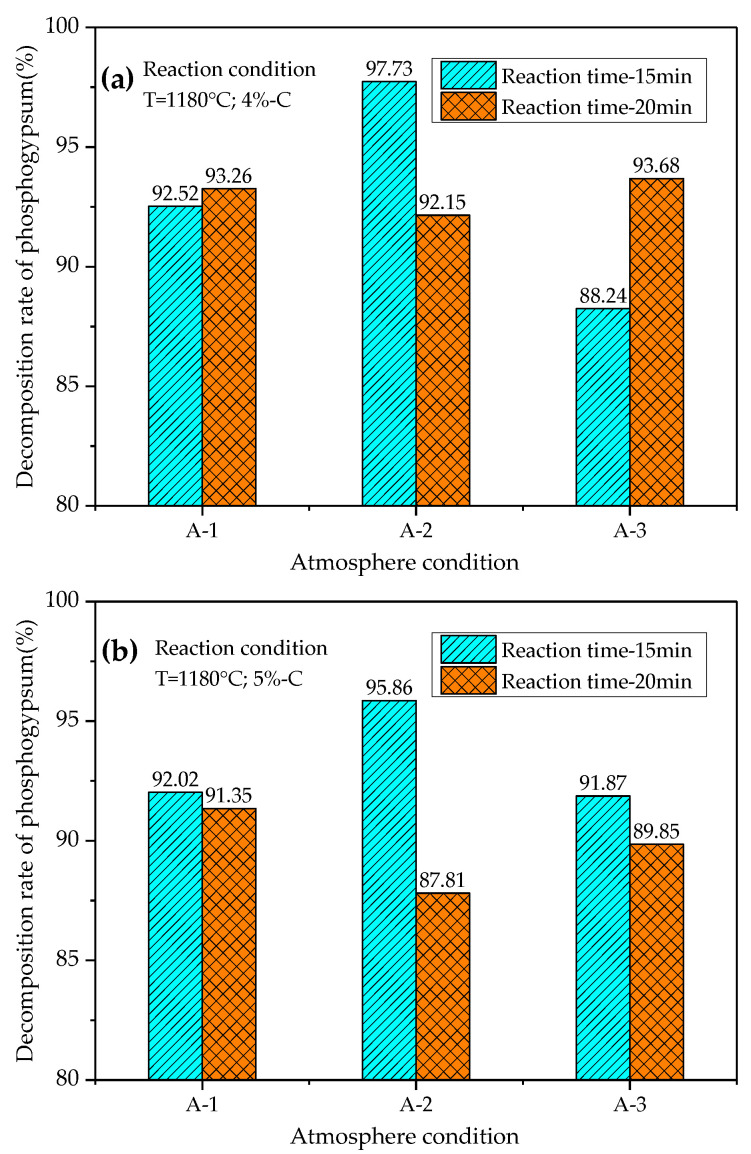
Decomposition rate (*ϕ*) of phosphogypsum under different atmosphere conditions: (**a**) the reaction temperature 1180 °C, time 15 min, carbon powder addition 4%; (**b**) the reaction temperature 1180 °C, time 20 min, carbon powder addition 5%.

**Figure 7 materials-14-01120-f007:**
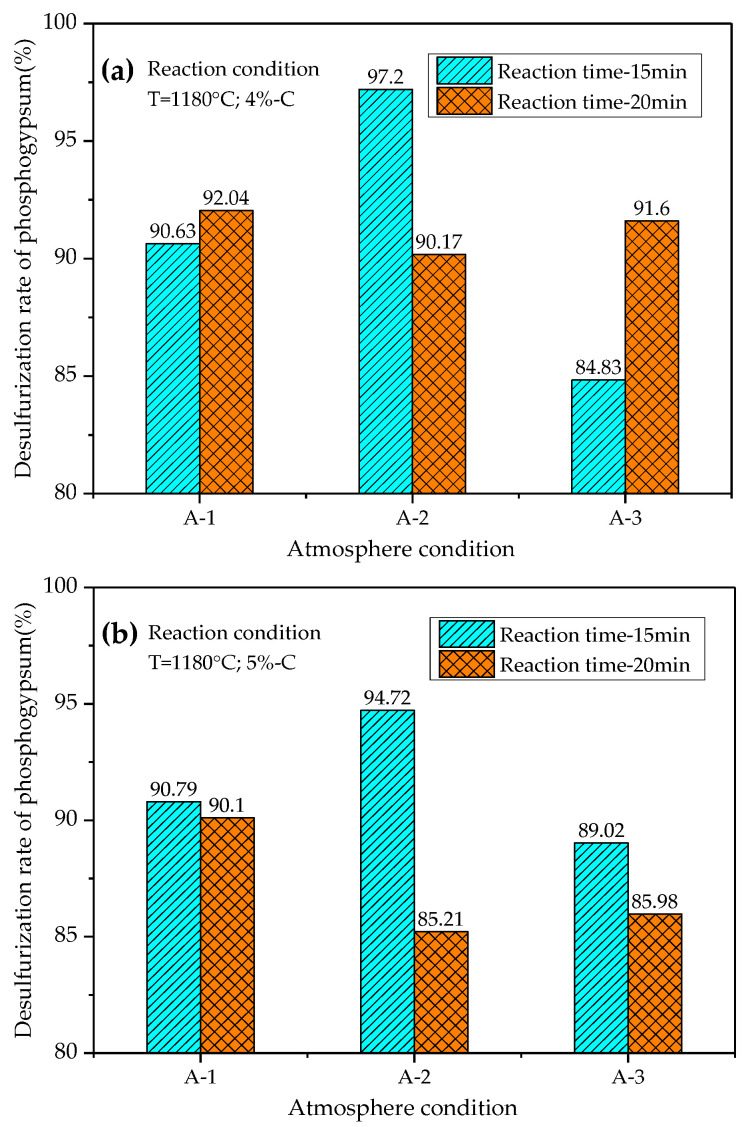
Desulfurization rate of phosphogypsum under different atmosphere conditions (*Ψ*): (**a**) the reaction temperature 1180 °C, time 15min, carbon powder addition 4%; (**b**) the reaction temperature 1180 °C, reaction time 20min, carbon powder addition 5%.

**Figure 8 materials-14-01120-f008:**
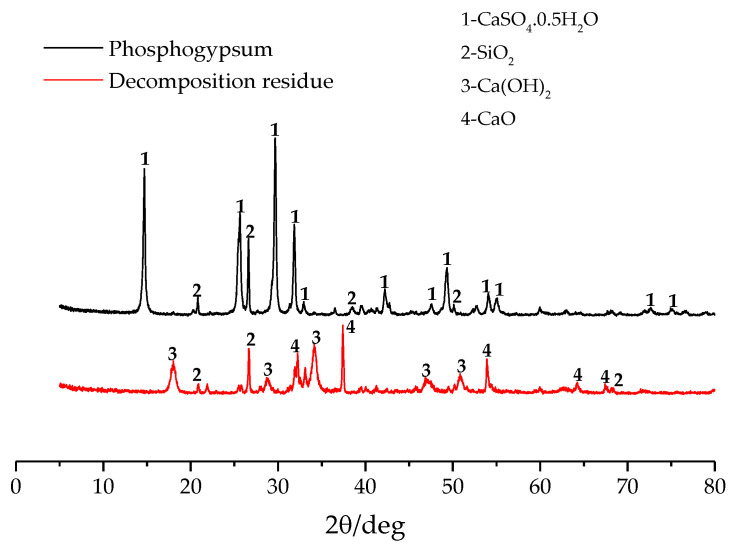
XRD of phosphogypsum decomposition residue.

**Figure 9 materials-14-01120-f009:**
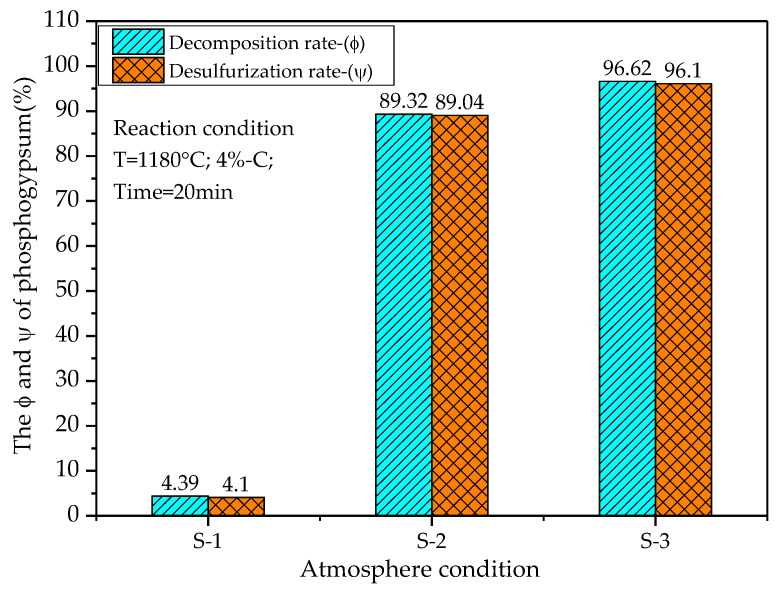
Effect of solid–gas and solid–solid reactions on phosphogypsum decomposition process.

**Figure 10 materials-14-01120-f010:**
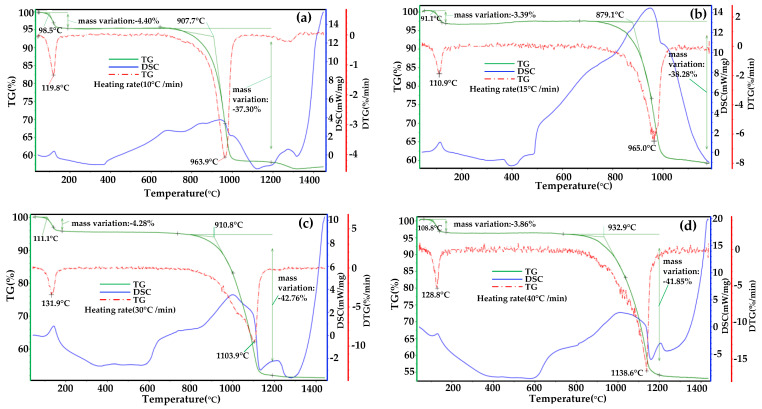
Analysis of decomposition heat of phosphogypsum at different temperature rising rates: (**a**) 10 °C/min; (**b**) 15 °C/min; (**c**) 30 °C/min; (**d**) 45 °C/min.

**Table 1 materials-14-01120-t001:** Chemical composition of the phosphogypsum (wt.%).

CaO	SO_3_	SiO_2_	P_2_O_5_	F	Al_2_O_3_	Fe_2_O_3_	MgO	Others
39.0	51.5	4.56	2.51	1.02	0.58	0.31	0.29	0.15

**Table 2 materials-14-01120-t002:** Starting temperature and free energy change of chemical reaction of phosphogypsum decomposition.

Reaction Number	Reaction Start Temperature (°C)	Free Energy $\Delta G_{T}^{0}$ (kJ)	Enthalpy of Reaction $\Delta H_{T}^{0}$ (kJ·mol^−1^)
1	1670	−3.594	860.402
2	930	−0.950	205.170
3	0	−171.302	−166.951
4	0	−437.243	−458.273
5	0	−866.050	−964.185
6	850	−0.572	584.447
7	460	−4.209	516.663
8	380	−3.861	346.261

**Table 3 materials-14-01120-t003:** Orthogonal test table.

NO.	Factor A: Temperature/°C	Factor B: Time/min	Factor C: Carbon Powder Content/%	Factor D: Atmosphere Condition
Level 1	1120	30	3.5	2
Level 2	1150	25	4	1
Level 3	1180	20	3	3

**Table 4 materials-14-01120-t004:** Atmosphere conditions.

Atmosphere Conditions	1	2	3
CO/%	4	6	8
CO_2_/%	22.3	27.3	30.7
N_2_/%	73.7	66.7	61.3
Pco/Pco_2_ (Partial pressure ratio of CO and CO_2_)	0.18	0.22	0.26

**Table 5 materials-14-01120-t005:** Orthogonal test results.

NO.	Factor A	Factor B	Factor C	Factor D	*ϕ*/%	*Ψ*/%
1	1120	30	3.5	2	76.51	64.50
2	1120	25	4	1	63.12	58.95
3	1120	20	3	3	55.70	44.87
4	1150	30	4	3	84.58	65.94
5	1150	25	3	2	77.30	67.69
6	1150	20	3.5	1	60.76	56.40
7	1180	30	3	1	77.43	68.61
8	1180	25	3.5	3	86.38	60.78
9	1180	20	4	2	79.33	67.04

**Table 6 materials-14-01120-t006:** Range analysis of orthogonal test.

**Name**	**Result**	**Factor A**	**Factor B**	**Factor C**	**Factor D**
Decomposition rate (*ϕ*)	K1	195.33	238.53	223.64	233.14
	K2	222.64	226.80	227.04	201.32
	K3	243.14	195.78	210.43	226.66
	R	47.81	42.75	16.61	31.82
	Superior Level	A3 (1180 °C)	B1 (30 min)	C2 (4%)	D1 (2)
	Order	A > B > D > C			
**Name**	**Result**	**Factor A**	**Factor B**	**Factor C**	**Factor D**
Desulfurization rate (*Ψ*)	K1	168.32	199.05	181.68	199.23
	K2	190.03	187.41	191.93	183.96
	K3	196.43	168.31	181.17	171.58
	R	28.11	30.74	10.76	27.65
	Superior Level	A3 (1180 °C)	B1 (30 min)	C2 (4%)	D1 (2)
	Order	B > A > D > C			

**Table 7 materials-14-01120-t007:** Atmosphere conditions.

Atmosphere Conditions	A-1	A-2	A-3
CO/%	5.00	6.00	7.00
CO_2_/%	27.78	30.00	31.82
N_2_/%	67.22	64.00	61.18
Pco/Pco_2_	0.18	0.20	0.22

**Table 8 materials-14-01120-t008:** Atmosphere conditions.

Atmosphere Conditions	S-1	S-2	S-3
CO/%	0	4	5
CO_2_/%	36	32	31
N_2_/%	64	64	64
Pco/Pco_2_	0.00	0.13	0.16

## Data Availability

The data presented in this study are available on request from the corresponding author. The data are not publicly available due to the requirements of the grant fund.

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
