# Peer review of "Experimental Study on Optimization of Phosphogypsum Suspension Decomposition Conditions under Double Catalysis"

_materials, 2021, doi:10.3390/ma14051120_

Round 1

Reviewer 1 Report

The manuscript reports on the optimization of catalytic thermal decomposition conditions of phosphogypsum, and their underlying effects. The paper is written well and presented in a clear manner, and hence can be considered for publication in Materials. The only suggestion is that when mentioning about building materials in the introduction, for the benefit of broader reach and to cover a wide range of building materials, the authors can consider adding a couple of additional references (pertaining to CSH) such as J. Phys. Chem. C, 2017, 121, 32, 17188–17196.

Author Response

Dear professor,

Reviewer 2 Report

Dear authors, your manuscript is very interesting, but some improvements are required  

General remarks: Why did you not investigate kinetics for reaction decomposition and reducing of phosphogypsum? You have all data for calculating the rate of these reactions and determine of mechanism. If you add kinetic calculations, this manuscript will get much better

Line 36:

You should add the link about the amount of phosphogypsum which formations during the phosphoric acid production process

Line 98 (Table 1):

It would be better if you will change of table name, for example on “Chemical composition of the phosphogypsum”.

Line 108 (2.2. Mechanism of Experiment):

It would help if you numbered of all chemical equations. For clearly understanding of the chemical mechanism, it would be better if you add a plot of dependence between change free energy and temperature (Ellingham diagram) for all chemical reactions. HSC Chemistry, FactSAGE or other thermodynamic software can be used for these calculations.

Line 187:

Why did you use fluorescence spectrometer for determine content of sulphur in residue? This method has a significant error in the analysis of light elements. It would be better to use other methods, such as infrared absorption of the gas method.

Line 214:

You should indicate the chemical equation for the pyrolysis process of phosphogypsum. Why is this process occurring at 1228 °C?

Line 221:

Why did you use 4% carbon? Is It a stoichiometric amount? You should indicate the reaction equation   

Author Response

Dear professor,

Reviewer 3 Report

Dear Authors:

The authors presented in their work a topic concerning the process of the decomposition reaction of phosphogypsum. Their work focused on the single reaction mechanism in gas-solid catalytic reaction or solid-solid catalytic reaction for optimization the ideal decomposition rate and desulfurization rate of phosphogypsum. It is important to find methods of utilization of waste phosphogypsum. However, different reaction conditions could get different products because of the complex components, which is the main factor of how the solid product is further used. Additionally, the authors used carbon as the traditional reductant for phosphogypsum decomposition. This process has been studied by Müler and Kühne. They discovered that coke could lower the decomposition temperature and increase the decomposition rate of gypsum. Industrial experimental of this coke process has been carried out in China since the 1950s.  

The experimental part of the article is well planned. However, the results presented in the paper are selected. It is difficult to sum them up well. Therefore, I believe that supplementing the research results and conducting another discussion is necessary to accept the article for publication in Materials.

Below, I present my comments on the work. 

Comment #1:

I think the "Introduction" and the discussion of the results can be improved by supplementing with publications on the decomposition of phosphogypsum using gas-solid and solid-solid reactions:

- Mihara, N.; Okumura, S.; Ozawa, S.; Kojima, Y.; Matsuda, H.; Kyaw Kyaw; Iwashita, T.; Goto, Y.; Ikehara, S.; Gushima, A.; Reductive decomposition of spent gypsum in CO—CO2—N2 and H2—CO2—N2 atmospheres, J Soc Inorg Mater Jpn 11  (2004) 266–271. doi.org/10.11451/mukimate2000.11.266

- Okumura, S.; Mihara, N.; Kamiya, K.; Ozawa, Sh.; Onyango, M. S.; Kojima, Y.; Matsuda, H.; Recovery of CaO by Reductive Decomposition of Spent Gypsum in a CO-CO2-N2 Atmosphere, Ind. Eng. Chem. Res. 42 (2003) 6046-6052. doi.org/10.1021/ie0302645

Comment #2:

Line 85: How was the particle size of phosphogypsum determined? If it has been determined by the authors, the method used should be specified in the text.

Comment #3:

Lines 91-93: Please specify how the particle size was calculated? You write about two ranges of size, the share of which is 62.51%. What about the rest? Remember that from the XRD results we get the crystallite size!

Comment #4:

Before you go to the descriptions in the experimental part, it is good to first describe what measurement methods and devices you used. In particular, I miss this description for Figures 1-3. SEM device is missing in the description below. What fluorescence spectrometer was used for sulfur analysis? Please check that all the devices you have used have been replaced together with the measurement parameters.

Comment #5:

Instead of placing in Fig. 3 two SEM micrographs at similar magnifications. List one at low magnification to see the difference in particle size.

Comment #6:

Separate each reaction on lines 121 and 127.

Comment #7:

Line 172: What was the process temperature?

Comment #8:

If you took equations 6 and 7 from the literature, give the reference.

Comment #9:

Line 221: You say 5% added carbon, while in figure 5b is 4%. Which version is correct?

Comment #10:

There is no information in the text on the parameters K1, K2, K3 presented in Table 5.

Comment #11:

In the description of Figure 6 you write: “Decomposition rate of phosphogypsum under different atmosphere conditions…”. However, I cannot find in the text how the atmosphere conditions changed? Perhaps you mean only a change in carbon? Please correct it.

Comment #12:

Line 305: The word “carbon” written in a different font.

Comment #13:

I do not fully understand the inclusion of detailed data from Analysis of decomposition heat of phosphogypsum at different temperature rising rates. The results of such analyzes are obvious. This can be shortened to a comment.

Comment #14:

Why is Table 8 included? The data from it are presented in Fig. 10. Decide on one form of data presentation.

Comment #15:

XRD results for all reactions performed are not shown in the text. For only one sample, the XRD results are presented in Fig. 8. For which decomposition reaction parameters is the XRD spectrum shown? The share of the individual components after decomposition is equally important. It is also good to use some comparative technique to analyze the composition, because XRD is not always able to detect compounds whose content is below 4 vol%. Such research is well presented in the publication:

Mihara, N.; Okumura, S.; Kojima, Y.; Matsuda, H.; Iwashita, T.; Goto, Y.; Ikehara S.; Gushima, A.; Regeneration of CaO from Spent Gypsum in CO and H2 reductive atmospheres, https://www.jstage.jst.go.jp/article/apcche/2004/0/2004_0_856/_pdf/-char/ja

Comment #16:

Please correct the discussion by referring to the information contained in the publications included in the review.

Author Response

Dear professor,

Round 2

Reviewer 3 Report

Dear Authors,

Thank you for all the answers to my questions. I accept the article in its present form.